# Alteration of Oral and Perioral Soft Tissue in Mice following Incisor Tooth Extraction

**DOI:** 10.3390/ijms23062987

**Published:** 2022-03-10

**Authors:** Takahiro Takagi, Masahito Yamamoto, Aki Sugano, Chiemi Kanehira, Kei Kitamura, Masateru Katayama, Katsuhiko Sakai, Masaki Sato, Shinichi Abe

**Affiliations:** 1Department of Anatomy, Tokyo Dental College, 2-9-18, Kandamisaki-cho, Chiyoda-ku, Tokyo 101-0061, Japan; takagitakahiro@tdc.ac.jp (T.T.); kanehirachiemi@tdc.ac.jp (C.K.); abesh@tdc.ac.jp (S.A.); 2Department of Dental Hygiene, Tokyo Dental Junior College, 2-9-18, Kandamisaki-cho, Chiyoda-ku, Tokyo 101-0061, Japan; asugano@tdc.ac.jp; 3Department of Histology and Developmental Biology, Tokyo Dental College, 2-9-18, Kandamisaki-cho, Chiyoda-ku, Tokyo 101-0061, Japan; kitamurakei@tdc.ac.jp; 4Department of Neurosurgery, Tokyo Dental College, Ichikawa General Hospital, 5-11-13, Sugano, Ichikawa 272-8513, Japan; mkatayama@tdc.ac.jp; 5Department of Oral Medicine and Hospital Dentistry, Tokyo Dental College, 5-11-13, Sugano, Ichikawa 272-8513, Japan; sakaik@tdc.ac.jp; 6Laboratory of Biology, Tokyo Dental College, 2-9-18, Kandamisaki-cho, Chiyoda-ku, Tokyo 101-0061, Japan; smsaki@tdc.ac.jp

**Keywords:** lip morphology, tooth loss, mouse, human, anatomy

## Abstract

Oral and perioral soft tissues cooperate with other oral and pharyngeal organs to facilitate mastication and swallowing. It is essential for these tissues to maintain their morphology for efficient function. Recently, it was reported that the morphology of oral and perioral soft tissue can be altered by aging or orthodontic treatment. However, it remains unclear whether tooth loss can alter these tissues’ morphology. This study examined whether tooth loss could alter lip morphology. First, an analysis of human anatomy suggested that tooth loss altered lip morphology. Next, a murine model of tooth loss was established by extracting an incisor; micro-computed tomography revealed that a new bone replaced the extraction socket. Body weight was significantly lower in the tooth loss (UH) group than in the non-extraction control (NH) group. The upper lip showed a greater degree of morphological variation in the UH group. Proteomic analysis and immunohistochemical staining of the upper lip illustrated that S100A8/9 expression was higher in the UH group, suggesting that tooth loss induced lip inflammation. Finally, soft-diet feeding improved lip deformity associated with tooth loss, but not inflammation. Therefore, soft-diet feeding is essential for preventing lip morphological changes after tooth loss.

## 1. Introduction

Tooth loss significantly impacts aesthetics, food selection, diet, and nutritional intake [1]; it additionally burdens people with severe mental illnesses [2], and increases the risks of cardiovascular disease [3,4,5], head and neck cancer [6], and esophageal cancer [7]. Because tooth loss reduces the volume in brain areas [8], it may be a risk factor for cognitive impairment and dementia [9,10,11]. Alveolar bone resorption occurs following tooth extraction, as a direct result of tooth loss [12]. However, it remains unclear whether tooth loss can alter the morphology of oral and perioral soft tissue.

The lips, which are oral and perioral soft tissue, are composed of surface epidermis, a muscle layer, connective tissue, and oral mucosal epithelium [13]. Lip morphology is altered in patients undergoing orthodontic treatment. Changes in the lip contour occur following maxillary incisor retraction [14]. The vertical position of the lip changes significantly following anterior tooth retraction [15]. Additionally, aging also affects lip morphology. Iblher et al. [16] reported that with age, the length of the upper lip increases, while its thickness decreases. Lip morphology or support is primarily derived from the cervical crown contour and alveolar ridge shape [17]. Alternatively, normal tooth and alveolar bone help maintain lip morphology. Thus, there is a need for improvement if tooth loss alters lip morphology.

In this study, we examined whether lip morphology was altered by anterior tooth loss. Moreover, we sought a means of improvement if tooth loss alters lip morphology.

## 2. Results

### 2.1. Morphology of the Upper Lip in Humans

MRI revealed that the lip bent toward the oral cavity side in group B compared with the findings in group A (Figure 1Aa–d). The angle between the α-line and the line connecting Su with In was measured (Figure 1Ae). The angle was smaller in group B (67.91° ± 9.97°) than in group A (90.45° ± 8.18°; Figure 1Af). The distance from Su to In, or N–Su, was not significantly different between the groups (Figure 1Ae,g,h). Similarly, in the cadaver analysis, the lip bent toward the oral cavity side in group B compared with the findings in group A (Figure 1Ba–c). The length of line b illustrated that the oral mucosa was more thickened in group B (1.33 ± 0.24 mm vs. 2.90 ± 0.97 mm; Figure 1Be). The lengths of line a and line c did not differ between the groups (Figure 1Bd,f). Therefore, lip morphology appeared to be associated with anterior tooth loss.

### 2.2. Incisor-Tooth-Deficient Mice Were Obtained by Extracting an Incisor Tooth

To demonstrate that tooth loss altered the lip morphology, mice lacking an incisor were generated (UH group; Figure 1A). Body size was smaller in the UH group than in the NH group (Figure 1B). Three to eight weeks after tooth extraction, body weight was significantly lower in UH mice than in NH mice (Figure 2B). Total protein levels were lower in UH mice than in NH mice (Figure 2B). New bone formation was identified in the extraction socket after eight weeks (Figure 2C). The lower incisor tooth in UH mice was extended compared with NH mice because the upper incisor tooth was extracted (Figure 2C). The width (b–d) of the area between the left and right lips was significantly different between NH (4.39 ± 0.46 mm) and UH mice (2.32 ± 0.41 mm), but there was no change in the major axis (a–c) of the area between the two (Figure 2D). Conversely, the width (b–d, b′–d′) and major axis (a–c) of the tongue did not differ between NH and UH mice (Figure 2E). Therefore, these results confirmed that the observed lip morphology alteration was attributable to tooth loss.

### 2.3. Measuring and Recording Lapping Behavior in Mice

Next, the oral function in the mice was measured following the protocol of Hayar et al. [18]. We recorded electrophysiological single units and tongue movement during lapping behavior. The lapping counts per second in the first half did not differ between NH and UH mice (Figure 3A); however, lapping counts in the latter half were smaller in UH mice than in NH mice (Figure 3A). Regarding lapping behavior, NH mice protruded their tongues anteroinferiorly, whereas UH mice protruded their tongues anterosuperiorly (Figure 3B).

### 2.4. Quantitative Proteome Analysis of Differences in Expression between NH and UH Mice’s Lips

In total, 1473 proteins were detected using high-sensitivity LC–MS/MS in the two groups. Among them, 1444 proteins displayed similar expression between the groups. However, 29 proteins were differentially expressed, with 15 exhibiting higher expression in UH mice and 14 displaying lower expression (Figure 4B). In the UH mouse lips, an enrichment of S100a8, S100a9, KRT6a, KRT6b, keratin 8, and keratin 16 expression was detected (Figure 4B).

### 2.5. An Upper Lip Comparison between Normal and Incisor-Tooth-Deficient Mice

Histological analysis of the upper lip also revealed a significant difference between the two groups. The distance from the a–b line to the tip of the lip was measured (Figure 5Ab–g) to analyze lip protrusion. The upper lip was more strongly protruded into the oral cavity in UH mice than in NH mice (Figure 5Ae–g). An incisor extraction increased the thickness of the upper lip’s epithelium (Figure 5Ba,g,m), and the cross-sectional area of the orbicularis oris muscle was reduced (Figure 5Bb,h,n). Because KRT6a, KRT6b, S100a8, and S100a9 expression was enriched in the UH mice’s lips (Figure 4), we then performed their immunohistochemical staining. KRT6A/6B fluorescence intensity in the granular layer was lower in UH mice than in NH mice (Figure 5Bc,d,i,j,o, and p; KRT6B of NH and UH data not shown). S100A8/A9 density was significantly higher in UH mice than in NH mice (Figure 5Be,f,k,l,q,r). These findings indicate that inflammation was induced by tooth loss, because S100A8/A9 is expressed during inflammation [19].

### 2.6. An Upper Lip Comparison between Hard-Diet and Soft-Diet Feeding in Incisor-Tooth-Deficient Mice

The effects of soft-diet feeding on lip morphology were assessed. Body weight was significantly higher in mice fed a soft diet than in those fed a hard diet (Figure 6Ab). Because the thickness and area of the lamina propria were smaller in mice fed a soft diet, the lip protrusion into the oral cavity was lower in these mice than in those fed a hard diet (Figure 6Ac–f). The muscle’s cross-sectional area (Figure 6Ba,f,k) and the epithelium’s thickness (Figure 6Ba,f,l) did not differ according to the diet. KRT6A/6B was more strongly expressed in the granular layer in mice fed a soft diet (Figure 6Bb,c,g,h). KRT6A/6B fluorescence intensity in the granular layer did not differ between mice fed a soft or hard diet (Figure 5Bp and Figure 6Bm,n; hard-diet feeding, KRT6B of UH data not shown). S100A8/A9 density did not differ according to the diet (Figure 6Bd,e,i,j,o,p). Mice fed a soft diet exhibited similar lip inflammation to those fed a hard diet, but soft-diet feeding improved lip protrusion into the oral cavity.

## 3. Discussion

Our previous study revealed the morphological association between the muscles and bones in the craniofacial region [20,21,22]. In this study, it was noted that the alteration of the tissue morphology affects the morphology of adjacent tissue. However, correlation does not prove causation. This study revealed that the alteration of the tissue morphology via tooth extraction affected adjacent tissue, namely, the lips. We believe that this phenomenon should be categorized as epigenetic, meaning that the lips did not involve underlying DNA sequence alteration [23,24,25]. DNA methylation, histone modification, and ncRNA interaction [26] may have occurred in the upper lips.

S100A8/A9 is a small calcium-binding protein belonging to the S100 family. Monocytes and neutrophils exhibit high S100A8/A9 expression, and accumulate in inflamed tissue following S100A8/A9 release [19]. Consequently, cytoskeletal rearrangement and arachidonic acid metabolism occur in inflamed areas. These findings revealed high S100A8/A9 expression in UH mice regardless of the diet, suggesting inflammation. This inflammation appeared to be caused by (1) the healing process of extraction sockets and/or (2) accidental contact of the lips with the tongue and hands. Due to anterior tooth loss, it is easy for mice to touch their upper lips with their tongue and hands. A dental prosthesis is essential to prevent lip inflammation.

KRT6, a type II intermediate filament protein, is expressed in the oral mucosa, tongue, esophagus, trachea, nail bed, hair follicles, palms, and soles [27]. KRT6 plays a vital role in wound healing [28] and mechanical support [27,29]. In the oral mucosa, KRT6 isoforms play an essential structural role in managing mechanical stress during suckling [30]. This study demonstrated that KRT6A/6B expression was higher in UH mice than in NH mice, which may be attributable to (1) the healing process of extraction sockets and/or (2) accidental contact of the lips with the tongue and hands.

There were no significant differences in body weight between the soft diet and the hard diet groups in normal mice [31,32]. *Mdx* mice lacking dystrophin—a protein that makes up the fibrous muscle membrane—have been used in various experiments [33,34]. Dystrophin deficiency necrotizes and reduces muscle fibers and muscle function, respectively; that is, soft-diet feeding leads to greater weight gain in dystrophic mice than hard-diet feeding [31]. The authors of [31] concluded that dystrophic mice cannot chew hard-diet food, and are thus underfed. In this study, body weight was significantly lower in UH mice than in NH mice. Low chewing ability may cause low-nutrient conditions. Conversely, body weight was significantly higher in US mice than in UH mice. Therefore, soft-diet feeding seems to revert body weight to its original status in incisor-tooth-deficient mice. With low chewing ability, hard-diet feeding is likely to harm the human body. Hyperglycemia causing long-term soft-diet feeding may elevate blood glucose, hypertension, and abnormal behavior [35]. When muscle function declines, muscle fiber composition changes [36,37,38,39,40]. Perhaps, in this experiment, it could be expected that the muscle fiber composition of the orbicularis oris muscle in the mice of the experimental group would changed to a more chicken-like type. However, we believe that this point should be clarified in future studies.

Lip inflammation is called cheilitis, which includes angular, contact, exfoliative, actinic, glandular, granulomatous, plasma cell, and simplex cheilitis, among others [41,42,43]. It is difficult to readily define the precise type of cheilitis [44]. Among them, contact cheilitis is attributed to the irritating or allergic effects of various substances found in many products; it usually presents as an eczema-like inflammation of the outer lip or vermilion margin, but extends to the surrounding skin and, less commonly, to the oral mucosa [45]. Our findings demonstrated that soft-diet feeding improved lip protrusion caused by an anterior tooth extraction. Soft-diet feeding may reduce mechanical stresses on the lips in incisor-tooth-deficient mice. The anterior tooth is likely to protect the lips from stress.

## 4. Materials and Methods

### 4.1. Magnetic Resonance Imaging (MRI) in Humans

Approval for a waiver of consent for retrospective chart review studies was granted by the ethics committee of Tokyo Dental College Ichikawa General Hospital (protocol #I 19-77R). A T2-weighted MRI (224 × 224 pixel matrix; 1.5 T, PHILIPS) was performed on 26 patients who visited the neurosurgery clinic of Tokyo Dental College Ichikawa General Hospital. The patients’ names were changed to case numbers to make individual identification impossible. The slice thickness was 3 mm, with an additional 0.3 mm gap. Patients were divided into two groups based on the presence of anterior teeth as follows: anterior teeth presence (group A), or anterior teeth absence (group B). The reference plane was defined as the line connecting the most depressed point of the nasal root with the pontomedullary junction (N–P line), as described by Okamoto (1989) [46] (Figure 1Ae). We plotted the subnasal point (Su) and the upper lip’s inferior tip (In). A line (α-line) that paralleled the N–P line and passed through Su was drawn (Figure 1Ae).

### 4.2. Gross and Histological Anatomy in Humans

This study was conducted following the provisions of the Declaration of Helsinki of 1995 (as revised in Edinburgh in 2000). Seven donated cadavers ranging in age from 60 to 95 years (mean: 83 years) were examined. The cause of death was ischemic heart failure or intracranial bleeding. These cadavers had been donated to Tokyo Dental College for research and education on human anatomy, and the university ethics committee approved their use for research (No. 932). The donated cadavers were fixed via arterial perfusion of 10% *v*/*v* formalin and stored in 50% *v*/*v* ethanol solution for >3 months. Cadavers were divided into two groups based on the presence of anterior teeth as follows: anterior teeth presence (group A) or anterior teeth absence (group B). We divided the upper lip on each cadaveric head into four tissue blocks (anterior side of the central incisor, lateral incisor, canine, and first premolar; Figure 1Ba). After performing routine procedures for paraffin-embedded histology, five serial sections were prepared and stained with Azan. The distances from the epidermis to the artery (skin side; line a), from the epithelium to the artery (oral side; line b), and from the tip of the lip to the artery (line c) were measured as described by Yamamoto et al. [13]

### 4.3. Generation of Incisor-Tooth-Deficient Mice

All experiments in mice were performed following the National Institutes of Health guidelines for the care and use of animals, and the Tokyo Dental College Institutional Animal Care and Use Committee approved the experiments (protocol #210102). We employed 42 C57BL6J mice (Sankyo Labo Service Corporation, Inc., Tokyo, Japan) at 15 weeks old. After a mixture of medetomidine (0.3 mg/kg), midazolam (4.0 mg/kg), and butorphanol (5.0 mg/kg) was administered to induce deep anesthesia, a murine model of incisor tooth loss was established by extracting an incisor tooth (Figure 2A). Weight was measured weekly after surgery to evaluate the animals’ health status. After confirming the loss of the incisor using micro-computed tomography (micro-CT), the mice were maintained for eight weeks (Figure 2B). Deep anesthesia was induced, blood was collected via cardiac puncture, and mice were finally euthanatized via CO_2_ intoxication. The upper lip and its surrounding tissue were harvested from each mouse, and various measurements were then performed.

### 4.4. Histological Analysis

The upper lip and its surrounding tissue were fixed in 4% phosphate-buffered paraformaldehyde. Specimens were decalcified using 10% ethylenediaminetetraacetic acid for 21 days at room temperature. Using standard methods, paraffin blocks were prepared, and a series of 5 μm thick tissue sections were cut using a sliding microtome (Leica Biosystems, Wetzlar, Germany). Then, the frontal sections were prepared, followed by staining with Masson’s trichrome.

### 4.5. Micro-CT Analysis

Normal mice (NH group) and mice lacking an incisor tooth (UH group) were compared to confirm the loss of the incisor. Imaging was performed using a micro-CT system (HMX 225Actis4; Tesco Co., Tokyo, Japan) under the following conditions: tube voltage, 100 kV; tube current, 120 μA; slice width, 50 μm; matrix size, 512 × 512; slice voxel size, 52.7 × 52.7 × 50 μm^3^. Images of the histological slides were used to reconstruct 3D images in VGStudio 3D reconstruction software (Volume Graphics, Heidelberg, Germany).

### 4.6. Lickometer and Recording of Licking Behavior

Mice were placed in a 14 × 14 × 7.6 cm^3^ plastic chamber with a 9 × 25 mm^2^ opening for tube access. The stainless steel tube was approximately 10 cm long. The chamber opening of the spout enabled contact of the spout with the tongue only during drinking. While recording licking behavior, the chamber’s bottom was covered with aluminum foil, and the water spout was connected to an A/D converter. The central pin of a BNC input connector of either a CED 1401 or a Digidata 1322A A/D converter was connected to the sipper tube, and the grounded housing (shield) of the BNC was connected to the aluminum foil. A 100–800 mV positive voltage step with rising times < 1 ms could be measured whenever each mouse’s tongue touched the steel sipper tube. The voltage signal did not require amplification, but it could be directly acquired using standard A/D interfaces. We used the powerCED 1401 (Cambridge Electronic Design, Cambridge, UK) with an input range of ±5 V and Digidata 1322A (Axon Instruments, Foster City, CA, USA) with an input range of ±10 V. Both interfaces had input resistances of 1 MΩ; they were equipped with 16 bit A/D converters, resulting in voltage resolutions of 0.15 (CED) and 0.3 mV (Digidata).

### 4.7. Quantitative Proteome Analysis

Six murine lip samples (three per group) were collected and stored at −80 °C until measurement. The frozen tissues were homogenized in 250 μL of protein extraction buffer (7 M urea, 0.1% NP-40, 500 mM TEAB) using grinding resin and a pestle. The lysate was sonicated (10 min, high level, 10 s on/off time interval) using the Bioruptor Sonicator (Diagenode, Liège, Belgium). The supernatant was collected after centrifugation at 18,800× *g* and 4 °C for 5 min. Then, 20% trichloroacetic acid was added to each sample to achieve a final concentration of 10%. The samples were vortexed, incubated on ice for 30 min, and centrifuged at 18,800× *g* and 4 °C for 30 min. Next, 500 μL of ice-cold acetone was added to the resulting pellet, vortexed, and then centrifuged at 18,800× *g* and 4 °C for 30 min. The pellet was washed twice with 500 μL of ice-cold acetone and resuspended in 40 μL of TMT lysis buffer (50 mM TEAB, 0.1% SDS). Protein concentrations were determined using the BCA method. A total of 15 μg of proteins per sample was reduced and alkylated as recommended by the TMT protocol. Samples were digested with trypsin (AB Sciex) overnight at 37 °C. TMT labeling was performed according to the manufacturer’s instructions of the TMTpro™ 16plex Label Reagent Set (Thermo Scientific (Waltham, MA, USA): Cat No. A44521). The labeled samples were combined and fractionated into six fractions using the ICAT strong cation-exchange column (AB Sciex), following the manufacturer’s instructions. Each fraction was concentrated via vacuum centrifugation and resuspended in 200 μL of 0.1% (*v*/*v*) formic acid. The samples were desalted using a MonoSpin^®^ C18 column (GL Science). The desalted samples were concentrated via vacuum centrifugation and resuspended in 40 μL of 0.1% (*v*/*v*) formic acid. All samples were stored at −20 °C until LC–MS analysis. Each peptide fraction was analyzed using Q Exactive Plus (Thermo Fisher Scientific (Waltham, MA, USA)) coupled online with a capillary high-performance liquid chromatography system (EASY-nLC 1200, Thermo Fisher Scientific) to acquire MS/MS spectra (Figure 4A). A 0.075 × 150 mm EASY-Spray column (3 μm particle diameter, 100 Å pore size, Thermo Fisher Scientific) with mobile phases of 0.1% formic acid and 0.1% formic acid/80% acetonitrile was used. Proteome Discoverer (version 2.4, Thermo Fisher Scientific) was used to search against the protein database SWISS-Prot and to quantitate TMT label-based quantification.

### 4.8. Immunohistochemical Analysis

The sections were incubated overnight at 4 °C with the following primary antibodies: mouse anti-keratin 6a (KRT6a) antibody (1:200 dilution; BioLegend, San Diego, CA, USA), rabbit anti-keratin 6b (KRT6b) antibody (1:200 dilution; Merck Millipore, Waltham, MA, USA), rabbit anti-S100A8 antibody (1:200 dilution; Abcam, Cambridge, UK), and rabbit anti-S100A9 antibody (1:200 dilution; Novus Biologicals, Minneapolis, MN, USA). Then, the sections were stained for 1.5 h at room temperature using the ABC staining kit (Funakoshi, Tokyo, Japan). Some sections were treated with ImmPACT 3,3-diaminobenzidine (Funakoshi, Tokyo, Japan) to detect any reaction, and were inspected after counterstaining with hematoxylin.

### 4.9. Statistical Analysis

All statistical analyses were performed using SPSS Statistics 21.0 (IBM, Armonk, NY, USA). *p*-values were calculated using Student’s *t*-test. *p* < 0.05 indicated significant differences between the groups (* *p* < 0.05; ** *p* < 0.01; *** *p* < 0.001 are used throughout the report). Error bars denote the standard deviations of the mean.

## 5. Conclusions

Assessments of human anatomy cannot prove causation, but we aimed to clarify the causes of changes in lip morphology following tooth extraction in animal experiments. Our findings demonstrated that the lip morphology was altered by tooth loss. More surprisingly, soft-diet feeding reduced lip protrusion into the oral cavity. Therefore, soft-diet feeding is essential for preventing the alteration of lip morphology after tooth loss.

## Figures and Tables

**Figure 1 ijms-23-02987-f001:**
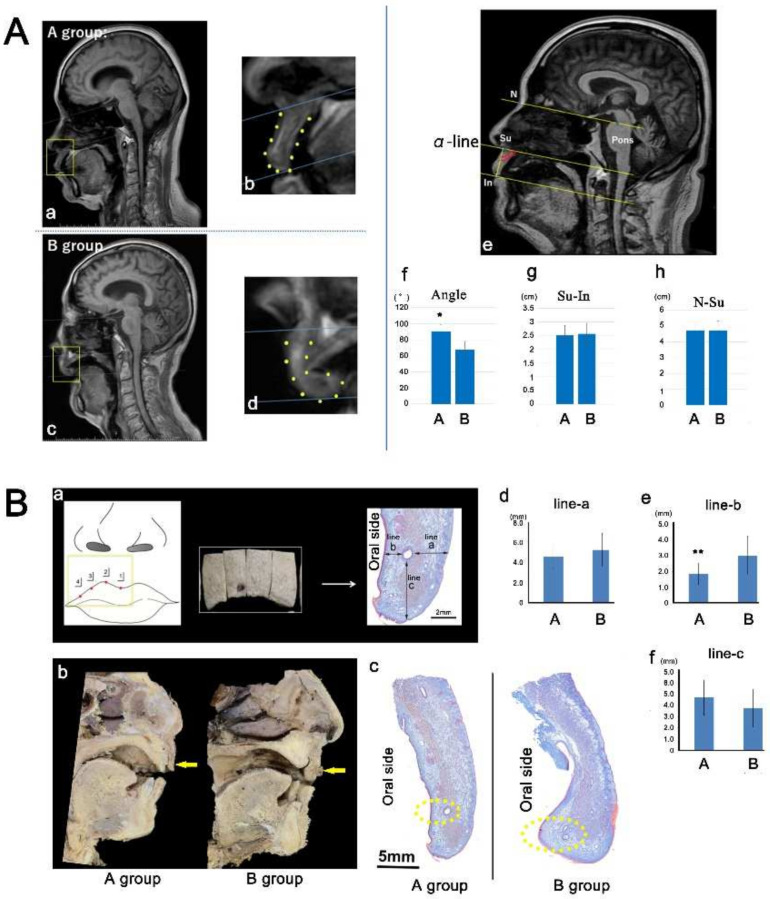
The morphology of the upper lip in humans: Patients and cadavers were divided into two groups based on the presence (Group A) or absence of anterior teeth (Group B). (**A**) Magnetic resonance imaging findings: The lip bent toward the oral cavity side in Group B compared with the findings in Group A (Panels a–d). The angle between the α-line and the Su–In line was smaller in Group B than in Group A (Panels e and f; * *p* < 0.05). The distance from Su to In, or N–Su, was not significantly different between the groups. (**B**) Cadaver findings: The lip bent toward the oral cavity side in Group B compared with the findings in Group A (Panels b and c). The length of line b revealed greater thickening of the oral mucosa in Group B than in Group A (Panels a and e; ** *p* < 0.01). The lengths of line a and line c did not differ between the groups (Panels a, d, and f). Su: subnasal point; In: inferior tip of the upper lip.

**Figure 2 ijms-23-02987-f002:**
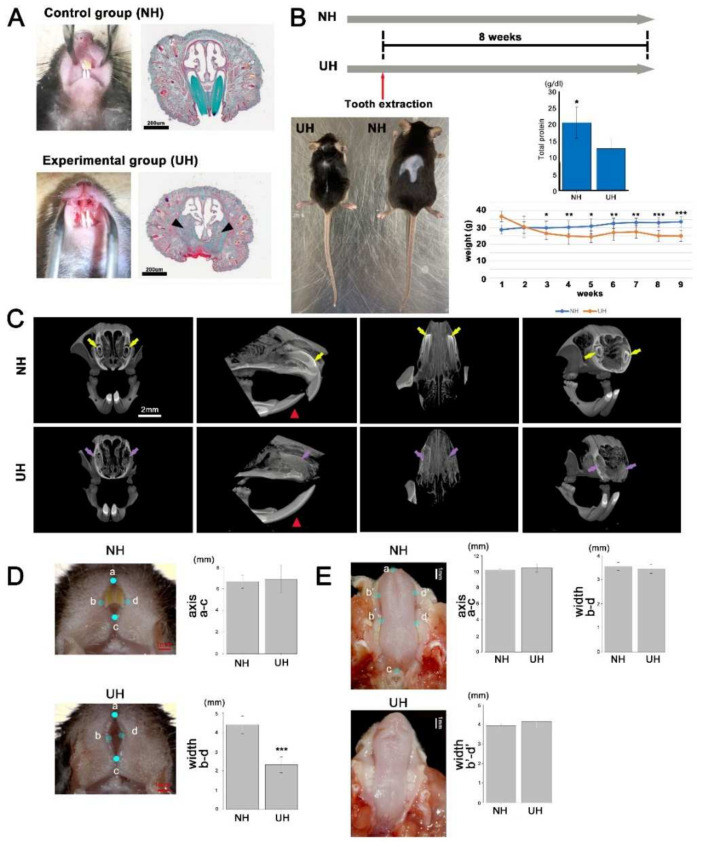
The establishment of incisor-deficient mice: (**A**) Before or after anterior tooth extraction. After tooth extraction, new bone formation was identified in the extraction sockets (arrowheads). (**B**) Physical description of the tooth loss (UH) and non-extraction control groups (NH). Body size was smaller in the UH group than in the NH group. Body weight was significantly lower in the UH group than in the NH group (* *p* < 0.05, ** *p* < 0.01, *** *p* < 0.01). The total protein expression was lower in the UH group than in the NH group (* *p* < 0.05). (**C**) Micro-computed tomography. Upper incisors were identified in the tooth socket (yellow arrows). New bone formation was identified in the extraction sockets (purple arrows). The lower incisor tooth was extended in UH mice compared with NH mice (red arrowheads). (**D**,**E**) Alteration of the lip and tongue morphology after tooth loss. The width of the area between the upper and lower lips was significantly different between the NH and UH groups (* *p* < 0.05), but there was no change in the major axis of the area between the groups. The width and major axis of the tongue did not differ between the groups.

**Figure 3 ijms-23-02987-f003:**
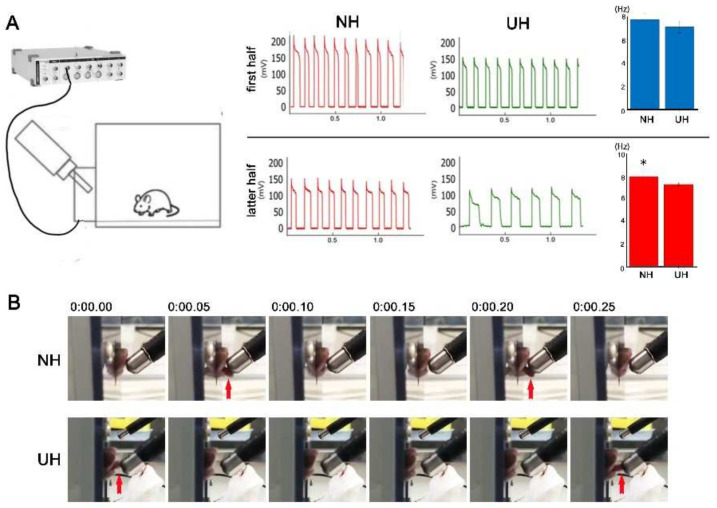
Lapping behavior in mice: (**A**) Electrophysiological single units during lapping behavior. Autocorrelogram of the lapping events revealed the temporally precise rhythmicity of the behavior at a periodicity of approximately 8 Hz. The upper row presents the first half, and the lower row presents the second half. Lapping counts in the second half were smaller in the tooth loss (UH) group than in the non-extraction control (NH) group (* *p* < 0.05). (**B**) Tongue movement during lapping behavior. Recorded tongue movements revealed that NH mice protruded their tongues anteroinferiorly, whereas UH mice protruded their tongues anterosuperiorly.

**Figure 4 ijms-23-02987-f004:**
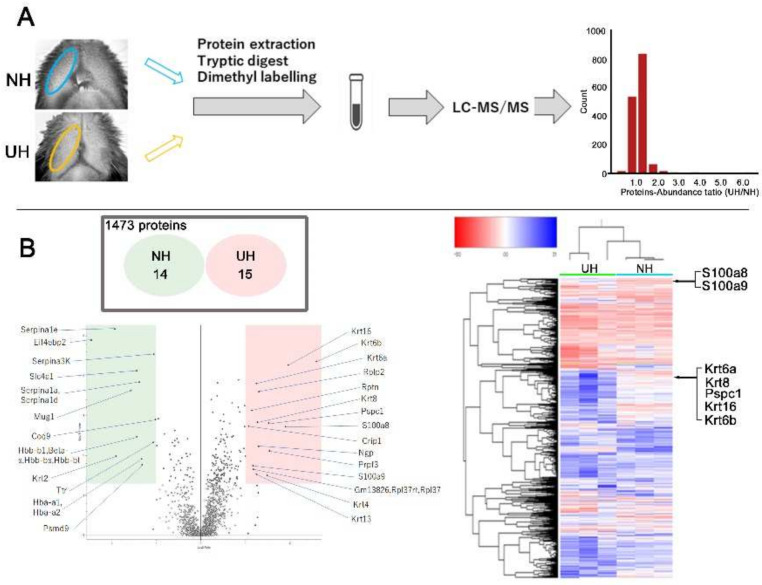
(**A**) Experimental procedure for the proteomic investigation. Histogram of protein counts grouped according to log2 protein ratios. In total, 1473 proteins were detected in the two groups using high-sensitivity liquid chromatography–tandem mass spectrometry. (**B**) Volcano plot of the statistical significance of differences (Student’s *t*-test *p*-value) as a function of the average protein ratios. Fifteen proteins were enriched in UH mouse lips (orange), while 14 were enriched in NH mouse lips (green). Heatmap illustrating quantitative alterations of representative proteins. Enrichment of S100a8, S100a9, keratin 6a (KRT6a), keratin 6b (KRT6b), keratin 8 (KRT8), and keratin 16 (KRT16) expression was detected in UH mouse lips.

**Figure 5 ijms-23-02987-f005:**
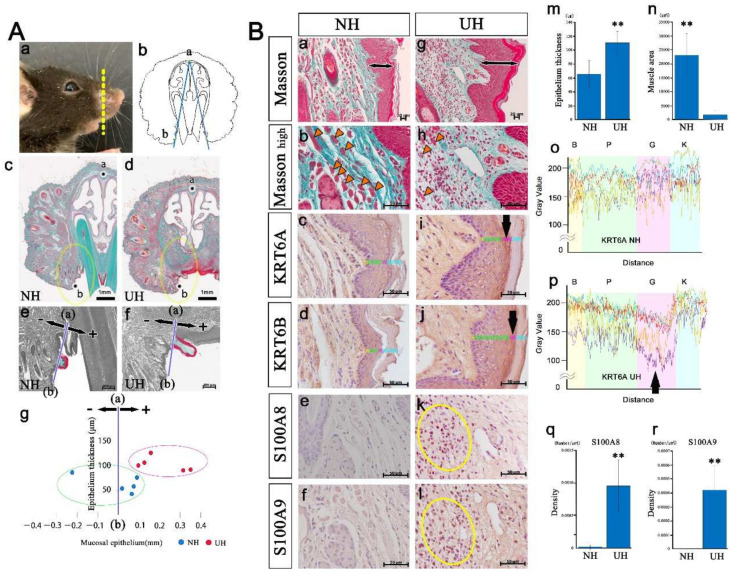
An upper lip comparison between the non-extraction control (NH) and tooth loss (UH) groups: (**A**) Histological studies of upper lips. The upper lip was more strongly protruded into the oral cavity in UH mice than in NH mice (Panels c–g). (**B**) High-magnification views of the upper lips. After extracting an incisor, the thickness of the upper lip’s epithelium was increased (Panels a and g) (Panel m; ** *p* < 0.01). The cross-sectional area of the orbicularis oris muscle (arrowheads) was reduced (Panels b and h) (Panel n; ** *p* < 0.01). KRT6A/6B fluorescence intensity in the granular layer was lower in UH mice than in NH mice (Panels c, d, i, j, o, and p). S100A8/A9 density was significantly higher in UH mice than in NH mice (Panels e, f, k, and l) (Panels q and r; ** *p* < 0.01).

**Figure 6 ijms-23-02987-f006:**
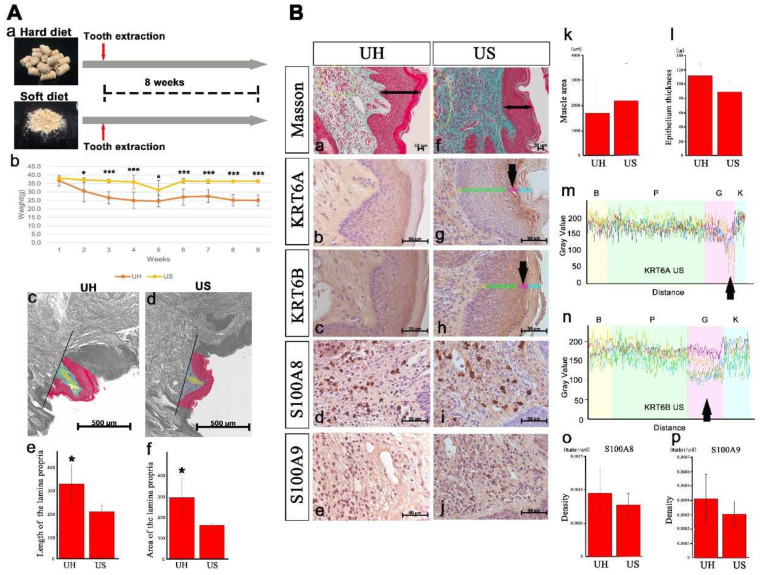
Soft-diet feeding affects lip morphology: (**A**) Body weight was significantly higher in the soft-diet feeding (US) group than in the hard-diet feeding (UH) group (Panel b; * *p* < 0.05, *** *p* < 0.001). Because the thickness and area of the lamina propria were smaller in US mice than in UH mice (Panels c and d) (Panels e and f; * *p* < 0.05), lip protrusion into the oral cavity was less severe in US mice. (**B**) The thickness of the epithelium and area of the muscle did not differ between the groups (Panels a, f, k, and l). The fluorescence intensity of KRT6A/6B in the granular layer did not differ between UH and US mice (Panels m and n). The density of S100A8/A9 was not different between the groups (Panels d, e, i, j, o, and p).

## Data Availability

Not applicable.

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
