# Peer review of "Alteration of Oral and Perioral Soft Tissue in Mice following Incisor Tooth Extraction"

_ijms, 2022, doi:10.3390/ijms23062987_

Round 1

Reviewer 1 Report

difficult topic but scientifically well supported and conducted with a rigorous method

Author Response

Does the introduction provide sufficient background and include all relevant references?

We removed the text included between lines 54 and 59 in introduction section and changed the purpose of this study.

Reviewer 2 Report

Dear authors

It is an interesting work presenting the oral and perioral tissue alteration after incisor tooth extraction in mice.

My suggestions are:

a. In Introduction section the text included between lines 54 and 59  should be removed because the results of the paper are presented in these lines.

b. The difference in body weight between groups must be discussed and justified and

c. a possible mechanism for reduced lip protrusion after incisor extraction in the soft diet group compared to the hard diet group must be discussed.

Author Response

a. In Introduction section the text included between lines 54 and 59  should be removed because the results of the paper are presented in these lines.

We removed the text included between lines 54 and 59.

b. The difference in body weight between groups must be discussed and justified and

In discussion section, we added a paragraph about the difference in body weight.

c. a possible mechanism for reduced lip protrusion after incisor extraction in the soft diet group compared to the hard diet group must be discussed.

In discussion section, we added a paragraph about a possible mechanism for reduced lip protrusion after incisor extraction in the soft diet group compared to the hard diet group.

d. English language and style are fine/minor spell check required.

We requested native speakers of English to proofread our English writing.

e.  Does the introduction provide sufficient background and include all relevant references?

We changed the purpose of this study.

Reviewer 3 Report

Dear Authors, I'd like to congratulate you for this comprehensive research about alteration in morphology of oral and perioral soft tissues following an upper anterior teeth loss.  The MRI human study and the histologic findings in cadavers demostrated alterations in lip morphology associated with anterior tooth loss. Moreover an animal model (mice with incisor defects) was studied in order to deepen knowledge from a histological, behavioural, Immunohistochemical point of view. 

Please check text editing: e.g."We believe this phenomenon should be categorized is epigenetic, meaning that the change in gene expression did not LINE 328"

Author Response

Please check text editing: e.g."We believe this phenomenon should be categorized is epigenetic, meaning that the change in gene expression did not LINE 328"

We change the text in line 328.